# Changes in Prolactin and Insulin Resistance in PCOS Patients Undergoing Metformin Treatment: A Retrospective Study

**DOI:** 10.3390/jcm13247781

**Published:** 2024-12-20

**Authors:** Tal Goldstein, Johannes Ott, Paula Katzensteiner, Robert Krysiak, Rodrig Marculescu, Magdalena Boegl, Marlene Hager

**Affiliations:** 1Klinik Favoriten, 1100 Vienna, Austria; talgold248@gmail.com; 2Clinical Division of Gynecological Endocrinology and Reproductive Medicine, Department of Obstetrics and Gynecology, Medical University of Vienna, 1090 Vienna, Austria; n11815961@students.meduniwien.ac.at (P.K.); magdalena.boegl@meduniwien.ac.at (M.B.); marlene.hager@meduniwien.ac.at (M.H.); 3Department of Internal Medicine and Clinical Pharmacology, Medical University of Silesia, 40-055 Katowice, Poland; rkrysiak@sum.edu.pl; 4Department of Laboratory Medicine, Medical University of Vienna, 1090 Vienna, Austria; rodrig.marculescu@meduniwien.ac.at

**Keywords:** prolactin, metformin, PCOS, insulin resistance, SHBG

## Abstract

**Background:** Prolactin levels have been shown to influence metabolic outcomes, including insulin resistance. Metformin is known to be beneficial in polycystic ovary syndrome (PCOS) patients. PCOS women might react differently to metformin treatment depending on their baseline prolactin levels. **Methods:** In this retrospective study, the homeostasis model assessment for insulin resistance (HOMA-IR), prolactin, luteinizing hormone (LH), follicle-stimulating hormone (FSH), the LH:FSH ratio, and total testosterone and sex hormone-binding globulin (SHBG) were measured in 75 obese/overweight women with PCOS and insulin resistance before initiation of metformin treatment and after 6–8 months. **Results:** At baseline, HOMA-IR was inversely correlated to SHBG (r = −0.408; *p* < 0.001) and prolactin (r = −0.402; *p* < 0.001). After 6–8 months of metformin treatment, the LH:FSH ratio and the HOMA-IR declined significantly (*p* < 0.05). A significant positive correlation could be shown between basal prolactin and the difference in the HOMA-IR (r = 0.233; *p* = 0.044). Women with lower baseline prolactin (≤14.9 ng/mL) revealed a sharper decline in HOMA-IR (−0.8, IQR −1.0; −0.5 vs. −0.6, IQR −0.8; −0.3; *p* = 0.049) as well as an increase in prolactin at follow-up (1.6 ng/mL, IQR −0.2;3.8 vs. −1.3, IQR −4.6;3.2; *p* = 0.003) compared to patients with a baseline prolactin > 14.9 ng/mL. **Conclusions:** In overweight/obese, insulin-resistant PCOS women, lower baseline prolactin levels are associated with higher baseline HOMA-IR levels as well as with a better response to metformin treatment. More data are necessary to prove these observations in larger populations.

## 1. Introduction

Polycystic ovary syndrome (PCOS) is among the most common endocrine diseases in women of reproductive age. The prevalence ranges from 6–20% and the manifestation of the disease is variable. The diagnosis is mostly based on the Rotterdam criteria, in which at least two out of the following three criteria have to be fulfilled: hyperandrogenism, either biochemical and/or clinical, polycystic ovarian morphology on ultrasound, with ≥20 follicles with a diameter up to 2–9 mm or an ovarian volume greater than 10 mL; and oligo- or anovulation [1,2].

Although the pathogenesis of PCOS is still not fully understood, metabolic aspects, which include insulin resistance, seem to play an important role [3]. It is believed that a post-receptor deficit causes insulin resistance in PCOS patients, which suggests an intracellular impairment [4].

In line with this aspect, metformin, a drug originally developed for diabetes, has also been used as a routine treatment in patients with PCOS in the last 20 years [4]. Metformin does not only work by lowering fatty acid oxidation and increasing glucose uptake in the periphery, but also by inhibiting hepatic gluconeogenesis and reducing the peripheral tissue’s sensitivity to insulin [5]. It has been shown that metformin can improve metabolic and hormonal profiles in patients with PCOS, such as fasting insulin, lipid parameters, testosterone, luteinizing hormone (LH)/follicle stimulating (FSH) ratio, sex hormone-binding globulin (SHBG), the glucose–insulin ratio, and prolactin [6]. Furthermore, metformin has been shown to improve weight reduction when combined with lifestyle modification [7], and can therefore also be used to induce ovulation [8].

In recent years, the literature demonstrated that high and very low prolactin levels were associated with negative metabolic outcomes, whereas moderately high prolactin levels were beneficial regarding various metabolic aspects, including insulin resistance [9,10,11,12], as reviewed recently by Macotela et al. [13]. Notably, prolactin seems to counteract insulin resistance and glucose intolerance, at least according to animal models, as recently reviewed by Macotela et al. [13]. In humans, a reduced prevalence of insulin resistance and protection from type 2 diabetes development have been found recently [13]. Based on the observation that prolactin exerts beneficial metabolic effects [13], one could assume that women with insulin-resistant PCOS might react differently to metformin treatment depending on their prolactin levels. Hypothetically, women with PCOS who also have higher prolactin levels might be more capable of a good treatment reaction to metformin. On the other hand, low prolactin levels could also be a sign of an insufficient reaction to the metabolic burden making metformin especially beneficial for patients with lower prolactin levels.

In a recent study about the effect of metformin on prolactin levels in PCOS women, Krysiak et al. included patients with hyperprolactinemia and pre-diabetes, with and without PCOS. Although metformin improved insulin sensitivity and increased SHBG in both study groups, the effect was stronger in patients without PCOS. Moreover, it was only the PCOS patients in whom metformin led to decreased prolactin levels [14].

Due to the small sample size of this study and general lack of longitudinal data about prolactin and metformin in women, we conducted a retrospective study and focused on pre-treatment prolactin levels and their association with outcomes after metformin treatment in women with insulin-resistant, overweight/obesity PCOS.

## 2. Materials and Methods

Study population: This retrospective observational study was conducted at the Clinical Division of Gynecologic Endocrinology and Reproductive Medicine of the Medical University of Vienna, Austria. This center is a referral center for gynecologic endocrinology and reproductive medicine. From January 2018 to December 2022, 75 women with PCOS according to the Rotterdam criteria [15] and insulin resistance, diagnosed as a homeostasis model assessment for insulin resistance (HOMA-IR) > 2.5, were included. The HOMA-IR was seen as a standard clinical evaluation tool since it was recommended by Teede et al. in the 2018 PCOS network recommendation [16]. Notably, it is not one hundred percent reliable. However, it has nevertheless shown high sensitivity and specificity in the PCOS population, when measured against the gold standard of insulin sensitivity testing, namely the clamp test [17]. In all these women, blood was drawn in a morning fasting state before initiation of metformin treatment between cycle days 2 and 5, resulting from either a normal cycle or bleeding after induction with dydrogesterone 10 mg twice daily for 10–12 days. Since the half-life of dydrogesterone and its active metabolite, 20a-hydroxydydrogesterone, is approximately 5–7 h and about 14–17 h, respectively [18], an influence on serum variables analyzed afterwards is unlikely. Furthermore, all these women had a body mass index (BMI) above 25 kg/m^2^ and underwent metformin treatment with a total daily dosage of 1500 mg–1700 mg, which was well tolerated by all of them. Moreover, all these women underwent complete laboratory follow-up after 6–8 months. Exclusion criteria were hypothyroidism, defined by a thyroid-stimulating hormone (TSH) above 4.0 µIU/mL, liver and kidney failure, and diabetes, diagnosed by HbA1c-values ≥ 6.5%. After 6–8 months of metformin treatment, blood was drawn again in a morning fasting state for hormone evaluation on cycle days 2–5, resulting from either a normal cycle or bleeding after induction with dydrogesterone as described above.

The study was conducted according to the guidelines of the Declaration of Helsinki, and approved by the ethics committee of the Medical University of Vienna (protocol code 2088/2023; date of approval: 1 February 2024).

Parameters analyzed: The AKIM software (SAP-based patient management system at the Medical University of Vienna; SAP GUI for Windows 7700.1.8.1161) was used for data acquisition. The main outcome parameters were serum levels of HOMA-IR and prolactin levels. In addition, serum levels of LH, FSH, LH:FSH ratio, total testosterone, and SHBG were also analyzed. Blood samples were obtained during the early follicular phase visit (cycle days 2–5). All serum parameters were determined at the Department of Laboratory Medicine, Medical University of Vienna, according to ISO 15189 quality standards [19]. As reported previously [20,21,22], Cobas electrochemiluminescence immunoassays (ECLIA) were performed on Cobas e 602 analyzers (Roche, Mannheim, Germany) for the determination of serum FSH, LH, testosterone, and SHBG. Prolactin was determined on Cobas e 801 ECLIA analyzers (intraassay variability ≤ 3.1%, interassay variability ≤ 3.8%). Moreover, the PCOS phenotype was also evaluated (A: hyperandrogenism + oligo-/anovulation + polycystic ovarian morphology; B: hyperandrogenism + oligo-/anovulation; C: hyperandrogenism + polycystic ovarian morphology; and D: oligo-/anovulation + polycystic ovarian morphology).

Statistical Analysis: Data are provided as median and interquartile range (IQR) for numerical parameters or number and frequency (%) for categorical parameters. Spearman rank tests were used for correlation analyses. Numerical parameters were compared using analyses of variances (ANOVA). The IBM Statistical Package for Social Science software (SPSS 28.0) was used for all statistical tests. *p*-values < 0.05 were considered significant. Levene’s test showed that many of the numerical parameters were distributed unequally.

## 3. Results

Basic patient characteristics and serum parameters before metformin treatment are presented in Table 1. There was approximately the same number of patients in each PCOS phenotype group, 20 patients in the group “phenotype A”, 19 patients in group “phenotype B” and “phenotype C”, and 17 patients in group “phenotype D”.

Concerning baseline prolactin levels and metabolic parameters, the following correlation analyses were performed (Figure 1): there were significant inverse correlations between HOMA-IR and SHBG (r = −0.408; *p* < 0.001) as well as between HOMA-IR and prolactin (r = −0.402; *p* < 0.001). In contrast, there was a significant positive correlation between SHBG and prolactin (r = 0.439; *p* < 0.001), whereas no significant correlation could be found between prolactin and BMI values (*p* = 0.113).

After six months of metformin treatment, the LH:FSH ratio and the HOMA-IR were significantly lower (*p* < 0.05), while there was no change in FSH, total testosterone, SHBG, and prolactin levels. Details can be seen in Table 2, which also provides an overview of the dynamics between baseline and 6–8 month follow-up levels.

Notably, when focusing on the median differences between baseline and follow-up levels (Figure 2), there were no significant correlations between HOMA-IR and prolactin (*p* = 0.703), nor between SHBG and prolactin (*p* = 0.837), nor between SHBG and HOMA-IR (*p* = 0.249). However, there was a significant positive correlation between the basal prolactin level and the difference in the HOMA-IR (r = 0.233; *p* = 0.044). In other words, the lower the initial prolactin level was, the more the HOMA-IR declined.

In a last step, patients were subdivided accordingly based on their initial prolactin level and the focus was on dynamics in prolactin and SHBG levels as well as HOMA-IR. The median difference between baseline and follow-up HOMA-IR was −0.8 (IQR −1.0; −0.5) for women with a baseline prolactin ≤ 14.9 ng/mL (n = 42) and −0.6 (IQR −0.8; −0.3) for women with a baseline prolactin > 14.9 ng/mL (n = 33; *p* = 0.049 between the two groups). Women with a baseline prolactin ≤ 14.9 ng/mL revealed a median increase in prolactin levels at follow-up (1.6 ng/mL, IQR −0.2; 3.8), whereas a decrease was found for the other group (median −1.3, IQR −4.6; 3.2; *p* = 0.003 between the two groups). No difference was found for the dynamics in SHBG (baseline prolactin ≤ 14.9 ng/mL: median 3.5 nmol/L, IQR 1.2; 5.9, versus baseline prolactin > 14.9 ng/mL: median 1.9 nmol/L, IQR −2.3; 4.6; *p* = 0.094). Table 3 provides an overview of baseline and follow-up levels in the two groups. Notably, in patients with baseline prolactin ≤ 14.9 ng/mL, prolactin levels increased significantly (*p* = 0.018), whereas HOMA-IR decreased (*p* < 0.001). In contrast, in the group of patients with baseline prolactin levels > 14.9 ng/mL, there was only a significant decrease in HOMA-IR (*p* < 0.001). Both groups revealed significant declines in the LH:FSH ratio (*p* < 0.05).

## 4. Discussion

This retrospective study, which included only patients with overweight/PCOS who met the criteria of insulin resistance based on HOMA-IR (also see Table 1), revealed several key findings. First, it confirmed the significant inverse correlation between HOMA-IR and prolactin levels in women with PCOS (r = −0.402; *p* < 0.001, Figure 1), which had already been mentioned previously for young women at risk for diabetes [23]. In addition, a significant negative correlation between HOMA-IR and SHBG was found (r = −0.408; *p* < 0.001), which is in line with previous findings in patients with PCOS [24,25,26] and likely due to hyperinsulinism inhibiting SHBG synthesis and secretion [27,28,29]. Given these results, it is reasonable that there was also a significant positive correlation between prolactin and SHBG (r = 0.439; *p* < 0.001).

A second major finding was the lack of significant correlations between the baseline to follow-up dynamics of prolactin and HOMA-IR (Figure 2). Neither the changes in HOMA-IR nor the changes in SHBG were associated with the dynamics in prolactin after 6–8 months of metformin treatment. However, metformin led to improvements in insulin sensitivity, which is supported by a significant decline in HOMA-IR and an increase in SHBG in our data set (Table 2). These findings are in line with the results published by Krysiak et al., where metformin also improved insulin sensitivity and increased SHBG but did not result in a decrease in prolactin levels in women with PCOS [14]. Similar results had been reported by other groups in small populations [30,31,32,33]. It has been mentioned that the lack of a decrease in prolactin after metformin might be due to higher testosterone levels [14], since similar results had been observed in men. In individuals with early-onset male-pattern hair loss, which might be the male PCOS equivalent [34], metformin produced a neutral effect on plasma prolactin [35], while hypogonadal men with prolactin excess revealed a decrease in prolactin only when exogenous testosterone was not given [36].

Lastly, in our study, we could see that the lower the initial prolactin level was, the more the HOMA-IR declined. This is somewhat different from the previous literature, which could be due to our lack of patients with very high prolactin levels in the study. Furthermore, women with lower baseline prolactin levels seemed to benefit more from metformin treatment in terms of HOMA-IR. However, these findings were only of moderate significance. In women with PCOS, an association of prolactin levels > 14.9 ng/mL with a decreased incidence of insulin resistance had been previously demonstrated according to a recent review [9]. Thus, in our study, we used the prolactin threshold of 14.9 ng/mL, which is in accordance with a previous review about the incidence of insulin resistance in patients with PCOS [9]. Given the assumption that moderately high prolactin levels would counteract insulin resistance and glucose intolerance [9], patients with lower baseline prolactin levels seem to be those who are incapable of a beneficial endocrine reaction to their abnormal metabolic status. It seems reasonable that these women benefit the most from medical treatment of insulin resistance, i.e., metformin in this case. Women with a baseline prolactin > 14.9 ng/mL revealed a decline in HOMA-IR only after 6–8 months of metformin treatment. In contrast, in women with a baseline prolactin ≤ 14.9 ng/mL, significant improvements in HOMA-IR were found alongside a significant increase in prolactin levels. Notably, the decline in HOMA-IR was greater in the group with a lower baseline prolactin (−0.8 versus −0.6, *p* = 0.049). Noteworthy, women with a lower baseline prolactin showed a significant prolactin increase at follow-up. One might hypothesize that the metformin-induced improvement in glucose metabolism was accompanied by a better capability to respond to metabolic issues with a rise in prolactin. Since Krysiak et al. included patients without PCOS with pre-diabetes who were hyperprolactinemic [14], it is hard to compare their results to ours. Without a doubt, future studies will have to focus on these issues in more detail.

Previous studies use very unconcise diagnostic criteria for both PCOS and hyperprolactinemia, which is why the exact prevalence of hyperprolactinemia in PCOS patients is still unclear [37]. There are various theories regarding the pathophysiological connection between PCOS and hyperprolactinemia. For example, there could be an abnormality in the hypothalamic–pituitary axis, which can cause PCOS and hyperprolactinemia at the same time. In fact, the release of prolactin and LH appears to be synchronized to a certain extent in PCOS patients [37,38,39,40]. LH secretion is slowed down by dopamine [37,41,42,43], which has led to the hypothesis that high LH levels may reduce dopaminergic tone and thus increase prolactin [37,42,44]. However, there is conflicting literature on this [37,45,46,47,48]. Furthermore, it is hypothesized that hyperprolactinemia could be triggered by the relative estrogen excess in PCOS [37,49,50]. Although some studies showed an increase in prolactin secretion under the influence of estrogen [37,51,52], other studies demonstrated that combined oral contraceptives (COC), for example, had no influence on the size of existing prolactinomas [37,53,54]. Another hypothesis is that there may be an enhanced GnRH pulsatility in PCOS women, which could lead to an increase in LH and thus to a negative influence on dopaminergic tone. This would then in turn lead to hyperprolactinemia [37,50]. However, there is no evidence of reduced prolactin levels in PCOS patients who previously received downregulation using GnRH agonists [37,55]. A real, scientifically proven pathophysiological connection between PCOS and hyperprolactinemia is still missing. Of note, concerning PCOS-specific parameters, there were no differences in the dynamics of testosterone and the LH:FSH ratio between the two groups. Apparently, patients with a lower baseline prolactin level revealed higher baseline testosterone levels (0.58 ng/mL versus 0.51 ng/mL), but this difference did not reach a statistical difference (*p* = 0.422). Thus, the baseline prolactin level does not seem to predict this specific outcome, at least after 6–8 months of metformin treatment.

The presented data need to be seen in the context of several study limitations: the retrospective study design, and the lack of data about other metabolic characteristics, first and foremost, the lipid profile. The sample size is also considerably small, especially given the well-known heterogeneity of PCOS with four phenotypes. Thus, larger and hopefully prospective data are needed to prove our findings. Moreover, for the last sub-analysis, we used a pre-defined prolactin cut-off value of 14.9 ng/mL in accordance with a previous review [9], which might not have necessarily been the optimized cut-off for the prediction of insulin resistance in our study population. Since we only included overweight/obese patients with insulin resistance, it was not possible to calculate a cut-off value in our study population. Furthermore, in our study, the PCOS phenotypes were equally divided, which they are usually not. This does not correspond to the distribution in the general population and should therefore also be regarded as a minor study limitation. Lastly, the HOMA-IR is a frequently applied tool for the assessment of insulin resistance and had been recommended as a standard tool in the 2018 international PCOS recommendation [16]. However, in a recent study about diagnostic accuracy, it revealed only a sensitivity of 50.9% and a specificity of 88.3% for insulin resistance diagnosed by the hyperinsulinemic, euglycemic clamp test in a large PCOS population [17]. Thus, the use of the HOMA-IR must also be seen as a minor study limitation. On the other hand, we consider the well-defined study population of women with overweight/obesity who underwent standardized metformin treatment for verified insulin resistance a strength.

## 5. Conclusions

In conclusion, in overweight/obese, insulin-resistant PCOS women, lower baseline prolactin levels are associated with higher baseline HOMA-IR levels as well as with a better response to metformin treatment. Accordingly, patients with lower baseline prolactin levels might benefit more from metformin treatment in terms of insulin resistance, although these associations were rather weak. More data are necessary to prove these new and interesting observations in larger patient populations.

## Figures and Tables

**Figure 1 jcm-13-07781-f001:**
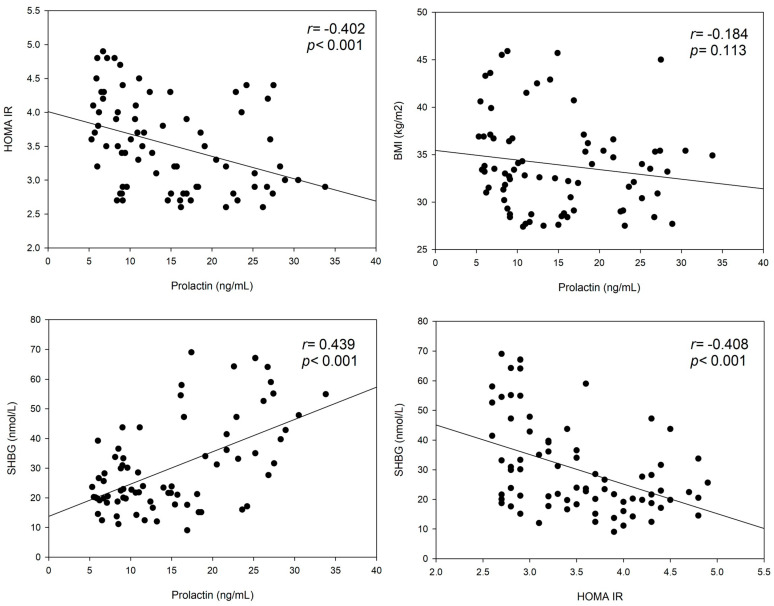
Correlations between metabolic parameters. Tested using Spearman rank correlation.

**Figure 2 jcm-13-07781-f002:**
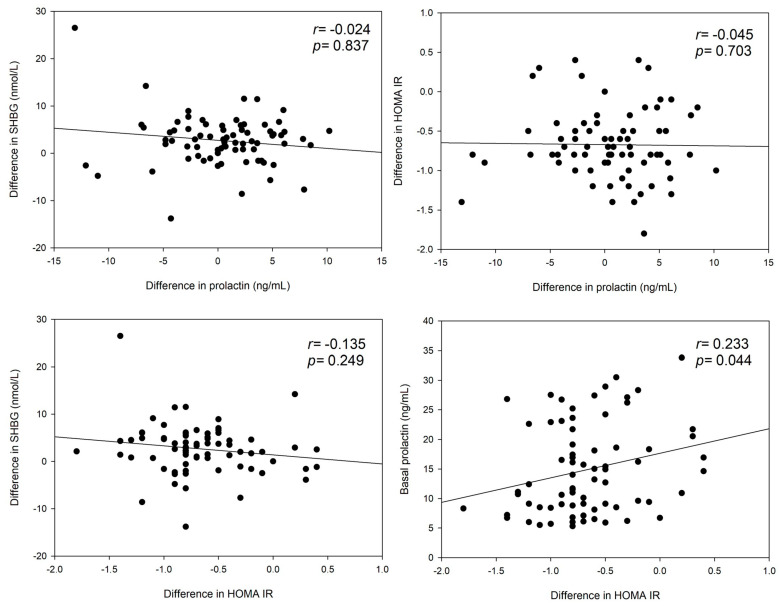
Pretreatment to follow-up dynamics of serum parameters: correlations.

**Table 1 jcm-13-07781-t001:** Basic patient characteristics and outcomes after 6–8 months of metformin treatment.

Age (years) ^1^		26 (23; 29)
BMI (kg/m^2^) ^1^		33.2 (30.2; 36.6)
PCOS phenotype ^2^	A	20 (26.7)
	B	19 (25.3)
	C	19 (25.3)
	D	17 (22.7)
LH (mIU/mL) ^1^		10.2 (7.7; 15.8)
FSH (mIU/mL) ^1^		5.6 (4.6; 6.3)
LH:FSH ratio ^1^		2.1 (1.4; 2.7)
Total testosterone (ng/mL) ^1^		0.54 (0.41; 0.69)
SHBG (nmol/L) ^1^		23.8 (19.1; 39.2)
HOMA IR ^1^		3.5 (2.9; 4.1)
Prolactin (ng/mL) ^1^		12.7 (8.5; 21.7)

Data are provided as ^1^ median (IQR) or ^2^ number (%). BMI = body mass index; PCOS = polycystic ovarian syndrome; LH = luteinizing hormone; FSH = follicle-stimulating hormone; SHBG = sexual hormone-binding globulin; HOMA IR = homeostasis model assessment for insulin resistance.

**Table 2 jcm-13-07781-t002:** Outcomes after 6–8 months of metformin treatment.

Parameter	Before Treatment	6–8 Months Follow-Up	*p*	Difference
LH (mIU/mL)	10.2 (7.7; 15.8)	9.7 (7.9; 12.8)	<0.001	−0.35 (−2.2; 0.7)
FSH (mIU/mL)	5.6 (4.6; 6.3)	5.8 (4.9; 6.8)	0.460	0.3 (−0.5; 1.3)
LH:FSH ratio	2.1 (1.4; 2.7)	1.8 (1.5; 2.1)	<0.001	−0.2 (−0.7; 0.1)
Total testosterone (ng/mL)	0.54 (0.41; 0.69)	0.48 (0.38; 0.62)	0.105	−0.04 (−0.08; 0.0)
SHBG (nmol/L)	23.8 (19.1; 39.2)	29.0 (21.6; 40.9)	0.264	2.8 (0.0; 5.1)
HOMA IR	3.5 (2.9; 4.1)	2.8 (2.3; 3.3)	<0.001	−0.8 (−0.9; −0.5)
Prolactin (ng/mL)	12.7 (8.5; 21.7)	13.7 (9.3; 19.5)	0.737	0.6 (−2.7; 3.6)

Data are provided as median (IQR). LH = luteinizing hormone; FSH = follicle-stimulating hormone; SHBG = sexual hormone-binding globuline; HOMA IR = homeostasis model assessment for insulin resistance.

**Table 3 jcm-13-07781-t003:** Dynamics in serum parameters according to the basal prolactin level.

		Prolactin ≤ 14.9 ng/mL (*n =* 42)	Prolactin > 14.9 ng/mL (*n =* 33)
Prolactin (ng/mL)	Before treatment	8.9 (6.7; 10.9)	22.6 (17.1; 26.8)
	Follow-up	10.3 (7.5; 14.4)	21.6 (13.9; 27.3)
	*p*	0.018	0.422
HOMA IR	Before treatment	3.8 (3.4; 4.3)	3.0 (2.8; 3.6)
	Follow-up	3.0 (2.5; 3.5)	2.6 (13.9; 27.3)
	*p*	<0.001	<0.001
SHBG (nmol/L)	Before treatment	21.7 (18.7; 28.3)	36.1 (21.1; 54.7)
	Follow-up	25.1 (19.8; 33.1)	39.8 (26.7; 55.5)
	*p*	0.058	0.717
LH:FSH ratio	Before treatment	2.1 (1.4; 2.7)	2.1 (1.5; 2.8)
	Follow-up	1.8 (1.5; 2.2)	1.7 (1.4; 2.2)
	*p*	<0.001	<0.001
Testosterone (ng/mL)	Before treatment	0.58 (0.45; 0.70)	0.51 (0.37; 0.64)
	Follow-up	0.51 (0.39; 0.63)	0.45 (0.37; 0.58)
	*p*	0.206	0.327

Data are provided as median (IQR). HOMA IR = homeostasis model assessment for insulin resistance; SHBG = sexual hormone-binding globulin; LH = luteinizing hormone; FSH = follicle-stimulating hormone.

## Data Availability

Data will be made available upon reasonable request.

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
