# Peer review of "Changes in Prolactin and Insulin Resistance in PCOS Patients Undergoing Metformin Treatment: A Retrospective Study"

_jcm, 2024, doi:10.3390/jcm13247781_

Round 1
Reviewer 1 Report
Comments and Suggestions for Authors
The paper entitled “Changes in prolactin and insulin resistance in PCOS-patients undergoing metformin-treatment: a retrospective study” is very interesting, and, in my opinion, worthy of publication in the Journal of Clinical Medicine. However, I have some suggestions for improvement:
1. The conclusions should be written more clearly and explained in greater detail. For ezample, the statement “…baseline HOMA-IR and prolactin levels are inversely correlated, which supports the metabolically protective effect of prolactin. Patients with lower baseline prolactin levels might benefit” is somewhat confusing.
2. Please specify which statistical tests were performed to decide on the use non-parametric tests.
3. Please include a refererence for the statement in lines 59-60: ”Notably, prolactin seems to counteract insulin resistance and glucose intolerance, at least according to animal models”
4. Ensure that abbreviations are explained in the footnotes of the tables.
5. The sentence “In women with PCOS, an association of prolactin levels >14.9 ng/mL with a decreased incidence of insulin resistance had been demonstrated previously according to a recent review [9].” should be moved to the Discussion section.
6. Please check the values in Table 2. the median value for HOMA-IR is the same (10.2), yet the p- value is reported as <0.001. It seems unusual to observe such a low p-value, when the medians are identical. This warrants further clarification, especially given that the concentration of SHBG is not statistically significant (p=0.264) despite the reported values being 23.8 versus 29.0 for the same group size.
7. Did the authors attempt to ensure similar percentages of PCOS phenotypes? Typically, phenotype A is the most prevalent, while phenotype D represents only 5-10% of all PCOS cases.
8. lines 155-156: the lower the initial prolactin level was, the more the HOMA-IR declined, while is line 56-57 is written: “high and very low prolactin levels were associated with negative metabolic outcomes, whereas moderately high prolactin levels were beneficial regarding various metabolic aspects, including insulin resistance” and than “HOMA-IR and prolactin levels are inversely correlated, which supports the metabolically protective effect of prolactin” or “ Patients with lower baseline prolactin levels might benefit more from metformin treatment in terms of insulin resistance”
9. Please remove unnecessary space in line 24.
10. Please correct: “isa” in line 82.
Author Response
Dear Editors,
Dear Reviewers,
We thank you fort the thorough review and assessment of our manuscript. We took care in revising it according to all recommendations. Please find a detailed point-by-point answer letter below. We believe that the revisions made to our manuscript helped us to improve its quality. We hope that it will be acceptable for the precious “Journal of Clinical Medicine” in its present form.
Kind regards,
Marlene Hager
Robert Krysiak
Johannes Ott
-- In the name of all authors --
The paper entitled “Changes in prolactin and insulin resistance in PCOS-patients undergoing metformin-treatment: a retrospective study” is very interesting, and, in my opinion, worthy of publication in the Journal of Clinical Medicine. However, I have some suggestions for improvement:
- The conclusions should be written more clearly and explained in greater detail. For example, the statement “…baseline HOMA-IR and prolactin levels are inversely correlated, which supports the metabolically protective effect of prolactin. Patients with lower baseline prolactin levels might benefit” is somewhat confusing.
Answer: This section has been changed. The following paragraph has been added (lines 304-309): “In conclusion, HOMA-IR and prolactin levels are inversely correlated in women who are either overweight or suffer from insulin-resistant PCOS. This supports the theory of prolactin’s metabolically protective effect. Patients with lower baseline prolactin levels might benefit more from metformin treatment in terms of insulin resistance, although these associations were rather weak. More data are necessary to prove these new and interesting observations in larger patient populations.”
- Please specify which statistical tests were performed to decide on the use non-parametric tests.
Answer: The following sentence was added to the methods section (line 128) “Levene´s test showed that many of the numerical parameters were distributed unequally.”
- Please include a reference for the statement in lines 59-60: ”Notably, prolactin seems to counteract insulin resistance and glucose intolerance, at least according to animal models”
Answer: The following reference was added:
- Macotela Y, Ruiz-Herrera X, Vázquez-Carrillo DI, Ramírez-Hernandez G, Martínez de la Escalera G, Clapp C. The beneficial metabolic actions of prolactin. Front Endocrinol (Lausanne). 2022 Sep 23;13:1001703
- Ensure that abbreviations are explained in the footnotes of the tables.
Answer: We apologize for this mistake. All abbreviations are explained in the footnotes of the revised tables.
- The sentence “In women with PCOS, an association of prolactin levels >14.9 ng/mL with a decreased incidence of insulin resistance had been demonstrated previously according to a recent review [9].” should be moved to the Discussion section.
Answer: The sentence was moved to the Discussion section (lines 236-238). There, the rationale for using this cut-off value for our analyses is also provided.
- Please check the values in Table 2. the median value for LH is the same (10.2), yet the p- value is reported as <0.001. It seems unusual to observe such a low p-value, when the medians are identical. This warrants further clarification, especially given that the concentration of SHBG is not statistically significant (p=0.264) despite the reported values being 23.8 versus 29.0 for the same group size.
Answer: We apologize for this typo. The median LH level at follow-up was 9.7 mIU/mL. We corrected it in table 2.
- Did the authors attempt to ensure similar percentages of PCOS phenotypes? Typically, phenotype A is the most prevalent, while phenotype D represents only 5-10% of all PCOS cases.
Answer: We did not attempt this. We agree with the reviewer and address the limitation in the revised manuscript as follows (lines 291-293): “Furthermore, in our study, the PCOS phenotypes were equally divided, which they are usually not. This does not correspond to the distribution in the general population and should therefore also be regarded as a minor study limitation.”
- lines 155-156: the lower the initial prolactin level was, the more the HOMA-IR declined, while is line 56-57 is written: “high and very low prolactin levels were associated with negative metabolic outcomes, whereas moderately high prolactin levels were beneficial regarding various metabolic aspects, including insulin resistance” and than “HOMA-IR and prolactin levels are inversely correlated, which supports the metabolically protective effect of prolactin” or “ Patients with lower baseline prolactin levels might benefit more from metformin treatment in terms of insulin resistance”
Answer: We apologize for the misleading wording. Lines 56-57 refer to the existing literature, while lines 155-156 (lines 173-174 in the revised manuscript) refer to our findings. The following sentence was included in the manuscript to make this difference better understandable: “Lastly, in our study, we could see that the lower the initial prolactin level was, the more the HOMA-IR declined. This is somewhat different from the previous literature, which could be due to our lack of patients with very high prolactin levels in the study.” (lines 231-233 in the revised manuscript)
- Please remove unnecessary space in line 24.
Answer: We assume that the space between “≤” and “14.9” was meant. We removed it, thank you!
- Please correct: “isa” in line 82.
Answer: Sorry – corrected!

Reviewer 2 Report
Comments and Suggestions for Authors
This study investigated the prolactin levels in relation to PCOS. The results and discussion are clearly presented. The following are suggested to improve this study:
1. The use of linear correlation (R) must be included in the statistical analysis.
2. Please include the clinical use of HOMA-IR and reference its sensitivity/specificity in the clinical diagnosis and management of PCOS. Include a more detailed discussion of the parameters used in HOMA-IR in the methods.
3. The R graphs are not linearly correlated (R < +/-0.5). Please explain how the P value remained significant and why Spearman's rank was utilized in the correlation analysis.
4. Please add more reference on the possible physiologic role of prolactin in PCOS and metabolism. Add any potential therapeutic targets for this mechanism.
Author Response
Dear Editors,
Dear Reviewers,
We thank you fort the thorough review and assessment of our manuscript. We took care in revising it according to all recommendations. Please find a detailed point-by-point answer letter below. We believe that the revisions made to our manuscript helped us to improve its quality. We hope that it will be acceptable for the precious “Journal of Clinical Medicine” in its present form.
Kind regards,
Marlene Hager
Robert Krysiak
Johannes Ott
-- In the name of all authors --
This study investigated the prolactin levels in relation to PCOS. The results and discussion are clearly presented. The following are suggested to improve this study:
- The use of linear correlation (R) must be included in the statistical analysis. à ups, stimmt: Correlations were evaluated using Spearman’s tests
Answer: Thank you, in the manuscript it is stated that “Spearman rank tests were used for correlation analyses.” (lines 124-125)
- Please include the clinical use of HOMA-IR and reference its sensitivity/specificity in the clinical diagnosis and management of PCOS. Include a more detailed discussion of the parameters used in HOMA-IR in the methods.
Answer: We thank the reviewer for this comment. The following sentence was added to the manuscript (lines 86-91): “The HOMA-IR was seen as a standard clinical evaluation tool since it was recommended by Teede et al. in the 2018 PCOS network recommendation. [16] Although it is not one hundred percent reliable, it has nevertheless shown high sensitivity and specificity in the PCOS collective, when measured against the clamp test, which is the gold standard of insulin sensitivity testing. [17]“
- The R graphs are not linearly correlated (R < +/-0.5). Please explain how the P value remained significant and why Spearman's rank was utilized in the correlation analysis.
Answer: The Spearman correlation was used to assess monotonic relationships between two continuous or ordinal variables. Variables in monotonic relationships tend to change together, but not necessarily at a constant rate of change.
- Please add more reference on the possible physiologic role of prolactin in PCOS and metabolism. Add any potential therapeutic targets for this mechanism.
Answer: We added the following paragraph to the Discussion section (lines 259-275):
There are various theories regarding the pathophysiological connection between PCOS and hyperprolactinemia. For example, there could be an abnormality in the hypothalamic-pituitary axis, which can cause PCOS and hyperprolactinemia at the same time. In fact, the release of prolactin and LH appears to be synchronized to a certain extent in PCOS patients [37,38,39]. LH secretion is slowed down by dopamine[40-42], which has led to the hypothesis that high LH levels may reduce dopaminergic tone and thus increase prolactin [41,43]. However, there is conflicting literature on this [44-47]. Furthermore, it is hypothesized that hyperprolactinemia could be triggered by the relative estrogen excess in PCOS [48,49]. Although there have been several studies showing an increase in prolactin secretion under the influence of oestrogen [50,51], other studies have shown that combined oral contraceptives (COC), for example, have no influence on the size of existing prolactinomas [52,53]. Another hypothesis is that there may be an enhanced GnRH pulsatility in PCOS women, which could lead to an increase in LH and thus to a negative influence on dopaminergic tone. This would then in turn lead to hyperprolactinemia [49]. However, there is no evidence of reduced prolactin levels in PCOS patients who previously received downregulation using GnRH agonists [54]. A real, scientifically proven pathophysiological connection between PCOS and hyperprolactinemia is still missing.

Round 2
Reviewer 2 Report
Comments and Suggestions for Authors
The changes improved the presentation of this paper. However, the following needs to be further changed prior to publication:
1. Please change the first sentence of the abstract that starts with "metformin." This takes away from the main objective of this paper.
2. Please clarify the conclusion and provide more detail than just stating "inverse correlation" between HOMA-IR and prolactin. Please state if increased prolactin is associated with lower HOMA-IR and better prognosis (?).
3. Some sentences are too long and hard to understand.
4. SHBG is sex hormone binding globulin.
Comments on the Quality of English LanguageEnglish construction should be improved.
Author Response
Dear Editors,
Dear Reviewer,
We thank you for the thorough review and assessment of our manuscript. We took care in revising it according to all recommendations. Please find a detailed point-by-point answer letter below. We believe that the revisions made to our manuscript helped us to improve its quality. We hope that it will be acceptable for the precious “Journal of Clinical Medicine” in its present form.
Kind regards,
Marlene Hager
Robert Krysiak
Johannes Ott
-- In the name of all authors --
Reviewer 2
- Please change the first sentence of the abstract that starts with "metformin." This takes away from the main objective of this paper.
Answer: Good idea! We have exchanged the first two sentences: “Prolactin levels have been shown to influence metabolic outcomes, including insulin resistance. Metformin is known to be beneficial in polycystic ovary syndrome (PCOS) patients.”
- Please clarify the conclusion and provide more detail than just stating "inverse correlation" between HOMA-IR and prolactin. Please state if increased prolactin is associated with lower HOMA-IR and better prognosis (?).
Answer: Thank you! We made the following changes:
Abstract, Conclusion: “In overweight/obese, insulin-resistant PCOS women, lower baseline prolactin levels are associated with higher baseline HOMA-IR levels as well as with a better response to metformin treatment.”
Main text, Conclusion: “In conclusion, in overweight/obese, insulin-resistant PCOS women, lower baseline prolactin levels are associated with higher baseline HOMA-IR levels as well as with a better response to metformin treatment. Accordingly, patients with lower baseline prolactin levels might benefit more from metformin treatment in terms of insulin resistance, although these associations were rather weak.”
- Some sentences are too long and hard to understand.
Answer: We agree. We changed the following long sentences:
- Original: “Although it is not one hundred percent reliable, it has nevertheless shown high sensitivity and specificity in the PCOS collective, when measured against the clamp test, which is the gold standard of in-sulin sensitivity testing. [17]”
Revised: “Notably, it is not one hundred percent reliable. However, it has nevertheless shown high sensitivity and specificity in the PCOS population, when measured against the gold standard of insulin sensitivity testing, namely the clamp test. [17]”
- Original: “Although there have been several studies showing an increase in prolactin secretion un-der the influence of oestrogen [36,50,51], other studies have shown that combined oral contraceptives (COC), for example, have no influence on the size of existing prolactinomas [36,52,53].”
Revised: “Although some studies showed an increase in prolactin secretion under the influence of estrogen [36,50,51], other studies demonstrated that combined oral contraceptives (COC), for example, had no influence on the size of existing prolactinomas [36,52,53].”
- Original: “Although there have been several studies showing an increase in prolactin secretion un-der the influence of oestrogen [36,50,51], other studies have shown that combined oral contraceptives (COC), for example, have no influence on the size of existing prolactinomas [36,52,53].”
Revised: “Although some studies showed an increase in prolactin secretion under the influence of estrogen [36,50,51], other studies demonstrated that combined oral contraceptives (COC), for example, had no influence on the size of existing prolactinomas [36,52,53].”
- Original: “. Whereas women with a baseline prolactin >14.9 ng/mL revealed a decline in HOMA-IR only after 6-8 months of metformin treatment, in women with a baseline prolactin ≤14.9 ng/mL, significant improvements in HOMA-IR were found alongside a significant in-crease in prolactin levels.”
Revised: “Women with a baseline prolactin >14.9 ng/mL revealed a decline in HOMA-IR only after 6-8 months of metformin treatment. In contrast, in women with a baseline prolactin ≤14.9 ng/mL, significant improvements in HOMA-IR were found alongside a significant in-crease in prolactin levels”
- SHBG is sex hormone binding globulin.
Answer: Sorry for this mistake, which could be found in the Abstract. Corrected!
